# SUBRead: Clustering Sub-Graphs for Graph-Level Readout

## Abstract

Graph Neural Networks (GNNs) have transformed graph representation learning tasks across domains from bioinformatics and social networks to engineering applications. In graph classification, the readout function is an important component of the GNN architecture as it aggregates node features into a compact graph-level representation. Standard readouts such as sum, mean, and max often fail on complex graphs, as they cannot capture structural dependencies or contextual relationships among nodes due to the over-compressive nature of these functions, which leads to information loss. To address these challenges, we propose SUBRead, an expressive readout function which integrates subgraph clustering with attention-based weighting to produce a graph-level representation that preserves local structural information while capturing global dependencies. SUBRead is fully differentiable and compatible with various GNN architectures. Experiments on bioinformatics and social network benchmarks demonstrate that SUBRead consistently outperforms existing readouts, improving accuracy and interpretability. We further evaluate SUBRead on a real-world automotive engineering problem, where the task is to classify vibration responses of structures, referred to as structural mode shapes, using attributed graphs derived from simulation results. Unlike common graph benchmarks where graphs vary in topology, the mode shape graphs share a similar topology but significantly differ in node features, making sub-graph essential and providing a unique benchmark for readouts. The analysis demonstrates that SUBRead not only outperforms existing readouts but also provides meaningful substructures comparable to expert reasoning.

## 1 Introduction

Graph-based representations have become essential for modeling complex systems such as social networks (Ju et al., 2024), molecules (Zhang et al., 2021), and geometric systems (Shi & Rajkumar, 2020). One of the fundamental tasks in this domain is graph classification, where the goal is to predict a label for an entire graph (Morris, 2022). Traditional approaches such as graph kernels (Kriege et al., 2020) construct graph-level representations using handcrafted similarity functions, but these methods are computationally expensive and less adaptable to task-specific data (Xie et al., 2022). In contrast, Graph Neural Networks (GNNs) provide an end-to-end framework that jointly learns high-level node features and generates graph-level embeddings for downstream tasks (Wu et al., 2020).

A GNN-based model for graph classification typically consists of two stages: (1) message passing, where node features are updated through neighborhood aggregation, and (2) graph-level readout, where node features are aggregated into a fixed-length vector and fed into a classification layer. While most prior research focused message-passing algorithms (Kipf & Welling, 2016; Veličković et al., 2017; Xu et al., 2018), the readout stage has received comparatively less attention (Baek et al., 2021). Commonly used readout functions such as sum, mean, and max pooling (Duvenaud et al., 2015; Xu et al., 2018) are flat operations that treat all nodes equally, which can limit their ability to capture discriminative features for downstream tasks (Ying et al., 2018). For example, in bioinformatics, different atom types contribute unequally to chemical properties, and capturing their interactions is critical. Similarly, in social network, group-level structures such as communities or tightly connected subgraphs often carry more discriminative information than individual nodes. In particular, classical methods such as graph kernels (Ralaivola et al., 2005; Zaheer et al., 2017;

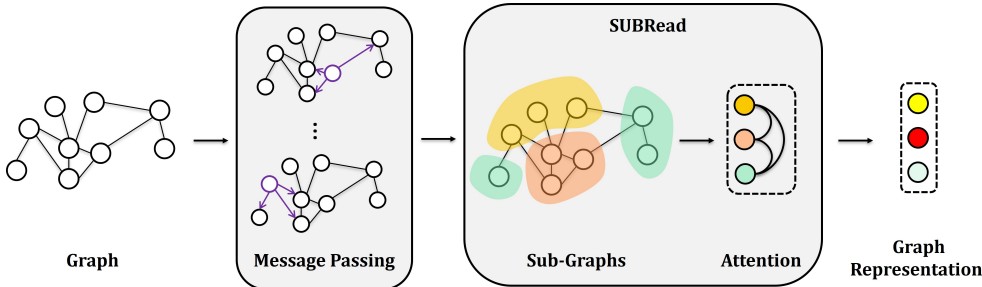

Figure 1: Overview of the SUBRead readout. The process begins after the message passing stage. The graph is first clustered into local sub-graphs, followed by an aggregation function that summarises each sub-graph into a vector. An attention mechanism is then applied among all sub-graph representation vectors to focus on the most relevant parts. The resulting graph-level representation is then fed into a classifier for a graph-level prediction.

Nikolentzos et al., 2021), which explicitly capture structural patterns through predefined similarity measures, have been shown to outperform GNN models on several benchmarks (Errica et al., 2020). This suggests that more expressive readout functions in GNN architectures are required to effectively capture higher-order dependencies in graphs.

To overcome the limitations of flat readout functions, researchers have proposed more expressive strategies for graph-level representations. Early efforts include Set2Set (Vinyals et al., 2015), adopted in MPNNs (Gilmer et al., 2017), which applies a recurrent attention mechanism to iteratively aggregate node features, and SortPool (Zhang et al., 2018), which constructs compact graph representations by sorting and truncating node embeddings. While these methods improve over flat pooling, Set2Set still treats the graph as an unordered set and lacks explicit structural bias, whereas SortPool enforces an artificial ranking of nodes that may discard important structural patterns. These shortcomings have motivated more recent approaches such as SOPool (Wang & Ji, 2020), which applies covariance-based pooling to capture higher-order feature statistics from all nodes, and Graph Multiset Transformer (GMT) (Baek et al., 2021), which improves graph readout by using learnable seed vectors that attend to different parts of the graph, allowing it to capture more diverse and informative representations. Despite these improvements, existing readout functions remain limited in their ability to explicitly capture subgraph-level information, which is often critical for distinguishing complex graphs.

In this paper, we introduce SUBRead, a novel readout function developed to enhance graph classification by capturing structural information through a combination of clustering and attention (as illustrated in Figure 1). In essence, the proposed method adopts a functional decomposition approach, where the graph-level representation is defined based on the modularity of its substructures. SUBRead consists of three steps: (1) the updated node features are grouped into subgraph-level representations through clustering, (2) an attention mechanism (Vaswani et al., 2017; Lee et al., 2019a) is applied to assign importance weights to these subgraph vectors, and (3) the weighted subgraph embeddings are concatenated to form the overall graph representation.

We experimentally validated the effectiveness of SUBRead on 11 benchmark datasets from bioinformatics and social networks (Morris et al., 2020) and found that our approach consistently outperforms existing readout functions. To further evaluate our approach, we introduce an additional dataset derived from a real-world automotive engineering problem, where the task is to classify vehicle structural vibration responses, known as mode shapes (Tohmuang et al., 2025), using attributed graphs constructed from simulation results (Appendix A.1). Unlike conventional benchmarks, where graph variation comes from both graph topology and node features. This dataset highlights the unique challenge of extracting discriminative sub-graph information, which SUBRead is designed to exploit. The problem setting is related to spatiotemporal graphs such as traffic networks (Jiang & Chen, 2023), but the structural mode shape graphs are static physical structures with varying node signals, making the task graph classification rather than node prediction. The results demonstrate that SUBRead not only outperforms existing readouts but also provides meaningful substructure-based decisions comparable to expert reasoning. Our main contributions are summarised as follows:

- We propose a novel readout function, SUBRead, which improves graph classification by aggregating sub-graph representations with attention-based weighting.

- We demonstrate that SUBRead consistently outperforms existing readouts on benchmark datasets from bioinformatics and social networks, when applied with GCN and GIN architectures.

- We apply SUBRead to a real-world vehicle structural vibration problem that emphasises the challenge of graph classification across highly similar but not identical topologies, and show that SUBRead produces substructures comparable to expert reasoning.

These contributions position SUBRead within the broader line of work on expressive readout functions and subgraph-aware GNN architectures. Whereas many prior approaches are tailored to specific domains or rely on substantial modifications to the backbone network and training objective, SUBRead focuses on the readout layer itself: it is a lightweight, architecture-agnostic module that can be plugged into standard GNNs without changing their message passing or loss. By evaluating SUBRead under a unified experimental protocol across bioinformatics, social-network, and engineering datasets, we provide a cross-domain perspective on when sub-structuring and attention are most beneficial for graph classification and how they compare to widely used readouts in practice.

## 2 RELATED WORK

### 2.1 GRAPH NEURAL NETWORK

Currently, there are several variants of GNN architectures, including Graph Convolutional Network (GCN) (Kipf & Welling, 2016; Xu et al., 2018; Veličković et al., 2017; Hamilton et al., 2017), Graph Recurrent Neural Network (GRN) (Ruiz et al., 2020; Li et al., 2015), Graph Transformer (GT) (Dwivedi & Bresson, 2020), and Graph Kernel Neural Network (GKNN) (Morris et al., 2019). While these approaches differ in their message-passing strategies, most have focused on node-level representation learning. In contrast, graph classification requires graph-level representations, typically obtained through a readout function that aggregates node embeddings into a single vector. Therefore, a key challenge in graph classification is to design a readout function that is expressive to distinguish different graph structures (Bianchi et al., 2020; Ju et al., 2024).

An influential line of work is the Graph Isomorphism Network (GIN) proposed by Xu et al. (Xu et al., 2018), which demonstrated that a sum operation yields strong theoretical expressiveness for graph classification. Despite this, subsequent studies reported that GIN can underperform on real-world datasets, in some cases even achieve lower classification accuracies compared to the earlier GCN model (Dwivedi et al., 2023). Moreover, GIN overlooks rich structural information, which has been shown to play a critical role in downstream prediction tasks (Zhang et al., 2019). These limitations have motivated research into more advanced architectures and readout strategies that can better capture whole-graph characteristics.

### 2.2 GRAPH READOUT & SETS OPERATIONS

In GNNs, the readout and pooling methods share similar operations, but serve different roles in the graph classification process. In this work, pooling refers specifically to hierarchical pooling, which progressively reduces the number of nodes between layers, similar to pooling in convolutional neural networks (Ying et al., 2018; Bianchi et al., 2020). In contrast, readout denotes the final aggregation of node embeddings into a single graph-level representation, performed only once after the last convolutional layer before classification. While hierarchical pooling can provide multi-resolution representations, studies have shown that hierarchical GNNs do not consistently outperform flat GNNs (a stack of convolutional layers without any node reduction), and that the convolutional layers themselves often play a more dominant role in determining the learned representations (Mesquita et al., 2020; Lee et al., 2019b). For this reason, our work focuses exclusively on the readout process.

Since readout functions disregard graph connectivity, the task can be formulated as set operations over node features. DeepSets Zaheer et al. (2017) provides the theoretical basis, showing that any permutation-invariant function can be decomposed into element-wise transformations followed by an invariant operation such as sum or mean. Set Transformer (Lee et al., 2019b) extends the concept

of DeepSets by replacing the simple aggregation function with an attention-based pooling architecture derived from the Transformer architecture for both encoding and decoding processes (Vaswani et al., 2017). Building on this idea, Graph Multiset Transformer (GMT) (Baek et al., 2021) adapts Set Transformer to graph readout by combining a message passing encoder with attention-based pooling. Unlike simple aggregators, GMT employs multiple learnable seed vectors to capture distinct information from the graph. However, because these seeds are not tied to specific regions of the graph, they may overlap in their attention, allowing some nodes to dominate the representation while others are neglected. To address this limitation, we propose a clustering-based readout that first partitions the graph into subgraphs and processes each group independently to focus on different parts of the structure. An attention mechanism is then applied over the resulting subgraph representations to model inter-subgraph dependencies, and the attended vectors are concatenated to form the final graph representation. This approach encourages more role-specific aggregation and enhances interpretability.

## 3 METHODOLOGY

### 3.1 PROBLEM FORMULATION

Given a set of labeled graphs $\mathbb{G} = \{(G_1, y_1), ..., (G_p, y_p)\}$ where $y_i$ is the label corresponding to the graph $G_i$. Each graph can be represented using adjacency matrix $A \in \{0, 1\}^{n \times n}$ and its node feature matrix $F \in \mathbb{R}^{n \times d}$, where $n$ and $d$ represent the number of nodes and features respectively. The objective is to predict a graph label for an unseen graph using a mapping function $f : G_{unseen} \rightarrow \hat{y}$.

### 3.2 GRAPH NEURAL NETWORKS FOR GRAPH CLASSIFICATION

The graph classification process consists of two steps. First, the model learns higher-level node representations by aggregating information from each node's neighborhood. Specifically, this process is formalised as Message-Passing function (Gilmer et al., 2017):

$$f_v^l = COMBINE^l(f_v^{l-1}, AGGREGATE^k(\{f_u^{l-1}, u \in N(v)\})) \tag{1}$$

where $AGGREGATE$ is the function applied to the neighbor nodes $N(v)$ at layer $l-1$, and then combined with the central node $v$ with the combination function ($COMBINE$). After $l$ layers, $f_v^l$ is the updated node feature at node $v$. Both $COMBINE$ and $AGGREGATE$ functions are developed in permutation-invariant fashion where the outputs remains constant regardless of the order of the nodes. The updated representations contain both structural and local features which will be utilised in various downstream tasks.

The second step is to transform the node feature vectors into a single fixed-length vector that serves as the representation of the entire graph. This process, referred to as $READOUT$, is applied after the final convolutional layer $L$ to produce the graph-level output for classification layer.

$$h_{G_i} = READOUT(f_v^L, v \in G_i) \tag{2}$$

where $h_{G_i}$ is a vector representation of graph $G_i$ after applying $READOUT$. The common functions are mean, sum, max or by more expressive approaches (Lee et al., 2019a; Zhang et al., 2018) over all node features $f_v^L$. The proposed $READOUT$ will be presented in the following section.

### 3.3 SUBREAD

Figure 1 illustrates SUBRead, which consists of two main stages. First, the graph is partitioned into multiple subgraphs. An aggregation function is then applied within each subgraph to obtain a summary vector, followed by a self-attention mechanism that models the relationships among these subgraph representations. Finally, the resulting vectors are concatenated into a single representation that encapsulates the entire graph. This concatenation depends only on the internal index of each learnable prototype, not on the ordering of nodes in the input graph: permuting the node order changes only the ordering of node embeddings, not the assignment of nodes to each prototype, so each sub-graph vector still aggregates over the same subset of nodes and permutation invariance

with respect to node permutations is preserved. We deliberately use concatenation rather than a permutation-invariant pooling across sub-graphs (e.g., sum or mean over $K$) so that different proto-types can learn distinct roles (for example, different functional groups or structural sub-assemblies) and remain distinguishable in the final representation; in this way, the $K$ sub-graph vectors act as $K$ fixed "slots" with potentially different semantics.

**Sub-Graphs:** Since GNNs update features only at the node level, producing node embeddings $H \in \mathbb{R}^{n \times d}$ (where $n$ is the number of nodes and $d$ is the feature dimension), the next step is to assign each node to a group based on its updated representation. One common approach is to use a set of induced vectors $C \in \mathbb{R}^{k \times d}$, where $k$ induced vectors acts as learnable prototypes that attend to the entire graph (Lee et al., 2019a; Baek et al., 2021). However, instead of relying on prototypes that aggregate information from all nodes which risks multiple prototypes collapsing onto the same subset, we propose to explicitly clusters nodes into subgraphs, ensuring that each group corresponds to a distinct local structure. To cluster the nodes, we compute a distance matrix between the node embeddings $H$ and the induced vectors $C$ using cosine distance, defined as follows:

$$\delta(h_n, c_p) = 1 - \frac{h_n^\top c_p}{\|h_n\|\|c_p\|} \tag{3}$$

where $h_n$ denotes the embedding of node $n$, and $c_p$ denotes the centroid of cluster $p$. Given a distance matrix representing pairwise distances between each node and each cluster, we can construct a sub-graph binary matrix, denoted as $\hat{C} \in \{0, 1\}^{n \times p}$. In this matrix, the entry in the column corresponding to the cluster with the minimum distance for a given node will be 1, while all other entries in that row will be 0. This formulation ensures that each node is associated with exactly one cluster, based on the minimum distance criterion.

Although this binary assignment resembles a classical $k$-nearest neighbours (kNN) rule, our formulation is conceptually closer to Neural Nearest Neighbours (N3Net) (Plötz & Roth, 2018). In N3Net, the non-differentiable kNN operator is replaced by a continuous relaxation using a temperature-controlled softmax over distances. Following this idea, we replace the hard winner–take–all assignment with an alignment matrix $\Pi \in \mathbb{R}^{N \times k}$ whose entries are

$$\pi_{np} = \frac{\exp\left(-\delta(h_n, c_p)/\tau\right)}{\sum_{p'=1}^{k} \exp\left(-\delta(h_n, c_{p'})/\tau\right)}, \tag{4}$$

where $n = 1, \dots, N$, $p = 1, \dots, k$, and $\tau > 0$ controls the sharpness of the distances. We therefore introduce an auxiliary alignment term $\mathcal{L}_{\text{align}} = \sum_{n=1}^{N} \sum_{p=1}^{k} \pi_{np} \delta(h_n, c_p)$ which is minimised with the supervised graph-level loss. As $\mathcal{L}_{\text{align}}$ decreases, strongly weighted pairs $(n, p)$ tend to exhibit smaller distances, promoting more coherent subgraphs.

**Attention Unit:** Based on the developed sub-graphs, we later apply an aggregation function to each sub-graph individually to obtain a subgraph-level representation. The choice of aggregation can vary depending on the downstream tasks. Note that flat aggregation within clusters is unlikely to cause significant data loss due to the similarity of the nodes within clusters. In this work, we use sum aggregation to compact each subgraph into a vector. After extracting a vector representation for each sub-graph, we then apply an attention function that learns to assign contributions of each sub-graph to the final graph-level representation. This allows the classifier to focus more on informative parts of the graph. Specifically, an attention function $Att(Q, K, V)$ maps a set of input vectors to a set of weighted output vectors which include interaction between sub-graphs. Given input vectors $X \in \mathbb{R}^{n \times d}$, where each row represents a sub-graph embedding, we compute queries $Q = XW^Q$, keys $K = XW^K$, and values $V = XW^V$, with learnable projection matrices $W^Q, W^K, W^V$. The standard attention function (Vaswani et al., 2017) is then applied as follows:

$$\text{Att}(Q, K, V) = \text{softmax}\left(\frac{QK^\top}{\sqrt{d_k}}\right) V$$

where $QK^\top \in \mathbb{R}^{n \times n}$ represents the pairwise similarity between each query and key vector. Each element $(QK^\top)_{ij}$ quantifies how much the $i$-th sub-graph (as a query) attends to the $j$-th sub-graph (as a key). This dot-product operation captures alignment or compatibility between sub-graph

Table 1: Classification performance for BioInformatic Benchmark

| GNN Architecture & Readout | PROTEINS | MUTAG | ENZYMES | NCI1 | Mutagenicity | D&D |
|---|---|---|---|---|---|---|
| GCN (Baseline) | 69.1 (6.0) | 73.4 (5.5) | 39.5 (4.3) | 70.6 (3.0) | 77.2 (1.5) | 74.7 (4.0) |
| GCN + SortPool (0.6) | 73.3 (4.4) | 76.5 (8.8) | 26.0 (5.2) | 74.4 (1.9) | 76.9 (1.9) | 77.6 (3.1) |
| GCN + Set2Set (p=2) | 73.8 (4.1) | 72.3 (5.7) | 48.8 (5.3) | 69.6 (2.7) | 80.0 (1.5) | 72.8 (3.9) |
| GCN + Set2Set (p=4) | 72.5 (4.8) | 72.8 (7.5) | 48.8 (6.4) | 69.2 (3.2) | 79.7 (1.6) | 72.8 (4.5) |
| GCN + Attention | 71.5 (3.4) | 72.3 (5.0) | 38.8 (6.0) | 70.1 (2.8) | 79.6 (2.2) | 68.8 (5.1) |
| GCN + Covariance | 73.3 (4.8) | 73.4 (6.8) | 44.5 (7.5) | 75.1 (2.0) | 78.2 (2.6) | 73.9 (4.4) |
| GCN + SOPool$_{Bimap}$ | 73.5 (3.9) | 73.4 (6.4) | 40.2 (8.5) | 73.6 (2.7) | 77.5 (2.1) | 74.0 (4.3) |
| GCN + SOPool$_{attention}$ | 65.8 (6.2) | 71.3 (5.5) | 22.3 (5.9) | 73.4 (2.9) | 76.5 (3.6) | 75.0 (3.3) |
| GCN + GMT | 71.3 (4.4) | 75.0 (6.7) | 39.7 (9.8) | 69.9 (2.4) | 79.3 (3.1) | 71.9 (4.9) |
| GCN + Janossy-GRU | 73.7 (3.0) | 77.8 (8.0) | 46.5 (7.0) | 75.0 (2.4) | 76.2 (2.2) | 71.5 (7.5) |
| GCN + SUBRead (k=2) | 74.4 (3.3) | 84.5 (5.6) | 39.5 (7.8) | 72.6 (4.1) | **80.4** (1.4) | **78.5** (3.3) |
| GCN + SUBRead (k=3) | 73.2 (1.8) | 80.0 (6.1) | 35.9 (9.2) | 75.3 (1.4) | 78.7 (0.6) | 76.9 (3.3) |
| GCN + SUBRead (k=4) | 74.5 (3.5) | **84.5** (7.7) | **50.7** (6.9) | **76.6 (1.5)** | 80.0 (1.7) | 77.2 (3.5) |
| GCN + SUBRead (k=5) | **74.8** (2.8) | 76.8 (7.9) | 35.3 (9.9) | 75.4 (1.0) | 76.6 (2.9) | 77.5 (4.0) |
| GIN (Baseline) | 74.6 (4.2) | 84.7 (8.6) | 48.7 (6.4) | 77.4 (2.1) | 81.2 (2.1) | 70.2 (3.8) |
| GIN + SortPool (0.6) | 73.9 (4.4) | 83.5 (8.0) | 48.7 (6.3) | 78.8 (1.3) | 79.4 (1.7) | 76.8 (3.3) |
| GIN + Set2Set (p=2) | 69.3 (4.0) | 83.5 (6.4) | 48.5 (7.9) | 76.1 (2.6) | 81.5 (2.0) | 71.3 (3.8) |
| GIN + Set2Set (p=4) | 72.1 (3.3) | 84.6 (8.9) | 33.8 (4.7) | 77.1 (1.5) | **82.1** (1.4) | 70.3 (4.6) |
| GIN + Attention | 71.0 (5.4) | 79.8 (10.6) | 41.5 (7.6) | 76.0 (2.8) | 81.3 (1.9) | 67.9 (5.0) |
| GIN + Covariance | 68.3 (7.2) | 81.9 (5.9) | 49.5 (5.1) | 77.4 (1.8) | 80.3 (1.4) | 61.3 (8.1) |
| GIN + SOPool$_{Bimap}$ | 69.1 (4.1) | 87.2 (4.9) | 45.8 (6.1) | 77.8 (1.5) | 81.3 (1.4) | 67.1 (5.1) |
| GIN + SOPool$_{Attention}$ | 69.4 (3.5) | 82.5 (6.2) | 44.2 (9.8) | 79.8 (1.1) | 80.5 (1.2) | 63.7 (4.9) |
| GIN + GMT | 71.0 (4.4) | 83.0 (7.6) | 44.2 (8.3) | 79.5 (1.8) | 81.8 (1.4) | 69.2 (4.2) |
| GIN + Janossy-GRU | 73.5 (4.0) | 83.0 (7.9) | 50.2 (8.1) | 77.9 (2.6) | 80.1 (2.5) | 67.9 (7.7) |
| GIN + SUBRead (k=2) | 73.3 (1.8) | **85.3** (9.1) | 55.3 (5.2) | 79.6 (3.0) | 80.6 (4.1) | 71.8 (3.7) |
| GIN + SUBRead (k=3) | **74.6** (2.9) | 82.1 (7.1) | 51.3 (2.2) | **80.3** (2.0) | 81.3 (1.3) | 70.4 (2.6) |
| GIN + SUBRead (k=4) | 71.9 (5.2) | 83.2 (7.7) | **59.0** (6.5) | 80.2 (2.0) | 81.7 (3.4) | **80.9 (2.5)** |
| GIN + SUBRead (k=5) | 72.3 (6.7) | 84.2 (7.4) | 53.3 (4.6) | 79.5 (2.4) | 80.0 (1.2) | 70.6 (4.8) |

embeddings. However, raw dot-product scores can be unbounded and hard to interpret. Therefore, we apply the *softmax* function across each row of $QK^\top$, converting these scores into between 0 and 1 with a scaling factor $\sqrt{d_k}$ for stable training. These weights determine how much each sub-graph should contribute to the final graph-level representation. To simplify analysis, we restrict our study to single-head attention. After obtaining the attention-weighted vectors for each sub-graph, they are concatenated into a 1D vector as a graph-level representation. This feature vector is subsequently fed into the classifier layer for graph-level prediction.

**Comparison with Subgraph-Based Message Passing Methods:** In parallel, several works incorporate subgraphs directly into the message-passing layers rather than only at readout. For example, subgraph GNNs treat a graph as a collection of rooted subgraphs and propagate information across these structures (Frasca et al., 2022), while motif-based attention uses higher-order motifs to weight messages between nodes (Lee et al., 2023). By contrast, SUBRead leaves the backbone unchanged and introduces subgraphs only at readout: after standard GNN layers, node embeddings are softly grouped into $K$ subgraphs and summarised with attention. This makes SUBRead architecture-agnostic and convenient to plug into existing GNNs, while still exploiting subgraph-level structure in the final representation. Conceptually, SUBRead is also related to virtual nodes, as each prototype can be viewed as a virtual node attending over all embeddings, but these prototypes operate solely at readout and do not modify the original graph or its message passing.

## 4 EXPERIMENTS

### 4.1 BENCHMARK DATASETS

1) **Datasets:** The experiments in this section were conducted on 11 benchmarks from the TU datasets collection (Morris et al., 2020). These include six datasets from bioinformatics (PROTEINS, MUTAG, ENZYMES, NCI1, Mutagenicity, D&D) and five datasets from social network domains (COLLAB, REDDIT-B, REDDIT-M, IMDB-B, IMDB-M). Since the social-network benchmark graphs do not have node features, we follow common practice (Ying et al., 2018; Xu et al., 2018; Errica et al., 2020) and use the node degree as the initial feature representation. The dataset statistics are provided in Appendix A.1.

Table 2: Classification performance for Social-media Benchmark

| GNN Architecture & Readout | COLLAB | REDDIT-B | REDDIT-MULTI | IMDB-B | IMDB-MULTI |
|---|---|---|---|---|---|
| GCN (Baseline) | 80.5 (1.4) | 82.8 (2.2) | 52.2 (2.2) | **75.2** (5.3) | **52.5** (1.8) |
| GCN + SortPool (0.6) | 81.2 (0.7) | 84.9 (3.2) | 50.1 (1.9) | 71.6 (3.9) | 51.2 (4.6) |
| GCN + Set2Set (p=2) | 80.5 (2.0) | 81.0 (6.6) | 51.4 (3.4) | 73.6 (2.9) | 49.9 (2.5) |
| GCN + Set2Set (p=4) | 80.5 (1.5) | 85.4 (0.9) | 53.4 (1.9) | 69.8 (5.0) | 50.7 (1.7) |
| GCN + Attention | 79.9 (1.5) | 87.9 (1.7) | 52.9 (3.2) | 74.1 (3.2) | 51.2 (2.8) |
| GCN + Covariance | 81.3 (1.9) | OOM | OOM | 73.9 (4.2) | 52.1 (2.8) |
| GCN + SOPool$_{Bimap}$ | 81.0 (1.3) | OOM | OOM | 73.3(3.6) | 51.8 (2.9) |
| GCN + SOPool$_{Attention}$ | 80.8 (1.0) | OOM | OOM | 73.2 (4.0) | 50.8 (1.9) |
| GCN + GMT | 79.3 (3.2) | OOM | OOM | 73.3 (3.2) | 47.7 (5.1) |
| GCN + Janossy-GRU | 79.5 (2.3) | OOM | OOM | 73.1 (4.4) | 50.3 (2.9) |
| GCN + SUBRead (k=2) | 81.5 (1.8) | 91.4 (1.1) | 53.9 (2.6) | 67.4 (5.2) | 51.7 (5.5) |
| GCN + SUBRead (k=3) | 81.2 (2.6) | 91.5 (1.8) | **55.1** (1.9) | 72.4 (3.8) | 52.0 (4.1) |
| GCN + SUBRead (k=4) | 82.4 (1.3) | **92.9** (1.9) | 54.9 (2.2) | 73.8 (2.0) | 50.4 (4.7) |
| GCN + SUBRead (k=5) | **82.5** (1.7) | 91.0 (1.7) | 53.8 (2.1) | 74.2 (2.3) | 48.9 (3.0) |
| GIN (Baseline) | 81.8 (1.0) | 78.1 (3.0) | 51.8 (1.9) | 74.0 (3.4) | 49.1 (2.4) |
| GIN + SortPool (0.6) | 82.2 (2.1) | 88.3 (1.8) | 50.6 (0.8) | 74.4 (2.6) | 52.0 (2.1) |
| GIN + Set2Set (p=2) | 82.8 (1.4) | 90.4 (2.4) | 53.1 (2.4) | **77.0** (5.7) | 49.9 (3.5) |
| GIN + Set2Set (p=4) | 80.6 (1.5) | **91.7** (1.8) | **53.4** (1.0) | 73.8 (5.6) | 50.9 (3.1) |
| GIN + Attention | 80.4 (2.3) | 87.6 (1.9) | 51.5 (3.3) | 72.8 (3.5) | 50.9 (3.4) |
| GIN + Covariance | 80.9 (1.6) | OOM | OOM | 72.2 (3.9) | 49.9 (2.3) |
| GIN + SOPool$_{Bimap}$ | 80.7 (1.9) | OOM | OOM | 73.8 (4.4) | 51.3 (3.1) |
| GIN + SOPool$_{Attention}$ | 77.3 (1.9) | OOM | OOM | 72.6 (4.1) | 49.5 (5.6) |
| GIN + GMT | 80.9 (1.7) | OOM | OOM | 73.1 (4.4) | 49.9 (2.3) |
| GIN + Janossy-GRU | 81.2 (2.3) | OOM | OOM | 72.1 (3.7) | 50.6 (5.8) |
| GIN + SUBRead (k=2) | **82.2** (2.1) | 83.2 (5.6) | 53.1 (2.7) | 71.8 (4.5) | 50.1 (4.8) |
| GIN + SUBRead (k=3) | 78.7 (3.0) | 79.0 (6.6) | 52.7 (2.3) | 70.4 (4.3) | 51.6 (3.8) |
| GIN + SUBRead (k=4) | 79.9 (2.8) | 82.0 (3.3) | 50.3 (1.9) | 70.4 (3.6) | **52.9** (3.7) |
| GIN + SUBRead (k=5) | 81.5 (1.1) | 81.0 (6.5) | 47.7 (5.2) | 71.8 (4.4) | 47.9 (2.0) |

OOM = out of memory

2) **Baseline:** We validate the effectiveness of SUBRead using two GNN architectures, Graph Convolutional Networks (GCN) (Kipf & Welling, 2016) and Graph Isomorphism Network (GIN) (Xu et al., 2018). Specifically, we compare SUBRead with existing readout functions, including Sort-Pool (Zhang et al., 2018), Set2Set (Vinyals et al., 2015), Attention (Girdhar & Ramanan, 2017; Li et al., 2019), Covariance (Wang et al., 2020), SOPool$_{Bimap\&Attention}$ (Wang & Ji, 2020), GMT (Baek et al., 2021), and Janossy-GRU (Buterez et al., 2022). These baselines were chosen because, similar to SUBRead, their primary objective is to construct a graph-level representation from the learned node features. For all baseline methods, we adopt the hyperparameter settings reported in the original publications for their best-performing configurations.

3) **Training Recipe:** For the quantitative evaluation of graph classification, we conduct a fair evaluation procedure suggested by (Errica et al., 2020). Specifically, we train each GNN classifier using 10-fold cross-validation, further splitting the training data into training and validation sets in a 9:1 ratio. Both GCN and GIN were implemented with three convolutional layers, followed by a readout and a classification layer. Training is conducted for a maximum of 500 epochs, with an early stopping criterion if the validation set performance does not improve for 50 consecutive epochs. We report the accuracy with mean values and standard deviations from all folds. For all experiments on TU datasets except D&D, the learning rate is set to $10^{-3}$, batch size to 128, hidden feature size to 128, weight decay to $10^{-4}$, and a dropout rate of 0.5 is applied at the classification layer. The temperature parameter is selected from the range $[0.1, 10]$ by choosing the value that yields the best validation performance. Since the D&D dataset has a large number of nodes (Table 6), the sub-graph methods can not perform on large graphs with large batch sizes, and therefore the hidden feature size and batch size are set to 64. The parameters in a GNN classifier are trained in an end-to-end manner by the supervised loss to learn discriminative graph features among the classes. Specifically for graph classification, the loss is defined based on the combination between $\mathcal{L}_{\text{align}}$ and Cross Entropy (CE), $L_{class} = -\sum_{i=1}^{N} y_i \log(\hat{y}_i)$, where $N$ is the number of classes , $y_i$ is the true label (0 or 1) for class $i$, and $\hat{y}_i$ is the predicted probability for class $i$.

4) **Results:** Table 1 reports the average classification accuracy and its standard deviation for the proposed method compared with existing works. The results show that SUBRead improves the performance of both GIN and GCN. A key observation from our experiments is that both models achieve higher performance when the number of induced vectors is set to 4. Specifically, induced vectors are important in summarising and transforming the graph's information, and their quantity directly impacts the model's ability to generalise and learn from the data. However, the variation in performance across different values of $k$ is generally small, except for small datasets such as MUTAG and ENZYMES, where the choice of $k$ has a more signficant impact.

As opposed to the bioinformatics datasets, Table 2 shows that SUBRead improves accuracy for the GCN architecture in the COLLAB, REDDIT-B, and REDDIT-MULTI datasets, and for the GIN architecture in the IMDB-MULTI dataset. These results suggest that the current sub-graph approach of SUBRead is particularly well adapted to bioinformatics datasets with clear physical substructures. For social network graphs, which are organised around communities and often exhibit more complex and irregular structures, SUBRead captures the data in a different way, helping the model focus on community-level interactions. While no additional gain is observed for IMDB-B dataset, the overall results still demonstrate the effectiveness of SUBRead across domains.

Table 3: Computational cost (wall-clock time per training) and model size (number of parameters) across datasets.

| Methods | PROTEINS | | NCI1 | | COLLAB | | REDDIT-B | |
|---|---|---|---|---|---|---|---|---|
| | Time (s) | #Params (K) | Time (s) | #Params (K) | Time (s) | #Params (K) | Time (s) | #Params (K) |
| GCN (Baseline) | 17.9 | 33.9 | 113.4 | 14.2 | 379.8 | 96.5 | 694.9 | 33.5 |
| GCN + SortPool (0.6) | 19.6 | 57.8 | 86.3 | 27.9 | 410.3 | 122.0 | 520.7 | 57.4 |
| GCN + Set2Set ($p$=2) | 31.7 | 231.8 | 106.1 | 64.0 | 546.5 | 294.5 | 359.8 | 231.4 |
| GCN + Set2Set ($p$=4) | 12.0 | 231.8 | 127.6 | 64.0 | 491.1 | 294.5 | 640.0 | 231.4 |
| GCN + Attention | 12.4 | 34.0 | 88.7 | 14.3 | 583.3 | 96.6 | 959.3 | 33.7 |
| GCN + SOPool (Covariance) | 74.3 | 66.4 | 97.3 | 22.3 | 926.2 | 145.3 | OOM | 66.0 |
| GCN + SOPool_Bi | 16.0 | 50.1 | 96.0 | 18.2 | 561.5 | 116.7 | OOM | 49.7 |
| GCN + SOPool_Attention | 68.7 | 34.0 | 99.0 | 14.3 | 597.1 | 96.6 | OOM | 33.7 |
| GCN + GMT (4) | 41.7 | 662.4 | 170.1 | 172.8 | 458.0 | 725.0 | 612.0 | 662.0 |
| Janossy-GRU | 140.7 | 149.5 | 665.6 | 43.3 | 622.4 | 212.1 | OOM | 149.1 |
| GCN + SUBRead (k=2) | 140.6 | 42.9 | 139.0 | 48.4 | 381.7 | 74.3 | 49.1 | 43.3 |
| GCN + SUBRead (k=3) | 22.3 | 43.2 | 92.5 | 48.6 | 364.2 | 74.6 | 113.2 | 43.0 |
| GCN + SUBRead (k=4) | 34.2 | 43.5 | 105.1 | 48.9 | 285.8 | 74.9 | 134.9 | 43.3 |
| GCN + SUBRead (k=5) | 32.4 | 43.7 | 131.6 | 49.2 | 1278.0 | 75.3 | 112.3 | 33.2 |

4) **Computational Complexity:** Table 3 reports the computational cost of each readout on four representative datasets (PROTEINS, D&D, REDDIT-BINARY, COLLAB), which span both bioinformatics and social networks and cover a wide range of graph sizes (see Table 6). For each method, we list the total wall-clock training time (seconds) for a single run (from random initialisation to early stopping or maximum epochs) and the number of trainable parameters (K). All experiments in this study were executed with an NVIDIA RTX A1000 GPU and 32 GB RAM. Overall, SUBRead maintains parameter counts in the same order as baseline GCN and attention-based models, and remain trainable on all datasets under the same computational resource.

## 4.2 VEHICLE MODE SHAPE CLASSIFICATION

1) **Datasets:** We also evaluate SUBRead on a dataset derived from a real-world engineering problem, in which the task is to classify structural vibration responses, known as mode shapes. This task is particularly important in the initial design stage, when structures are developed as Finite Element (FE) models and undergo several analyses and modifications before a physical prototype is built. Currently, manual labelling of mode shapes for each design is labour-intensive and inconsistent, which motivates the need for automatic classification. Because FE models are defined by nodes and their connectivity, they can be directly represented as graphs, where node features encode modal displacements and edges reflect structural relationships. This makes graph-based learning methods especially suitable for mode shape classification.

Unlike conventional benchmarks, where variations in both graph topology and node features provide strong cues for classification, the graphs in this dataset share highly similar, but not identical topologies because they originate from FE models of related structures. This makes sub-graphs reasoning for classification more critical, as different parts of the structure may contribute differently

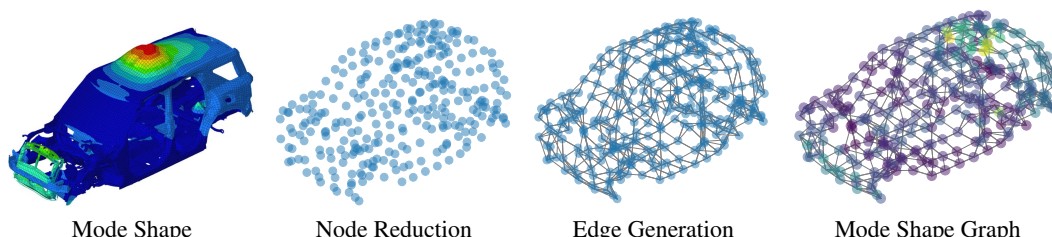

Mode Shape        Node Reduction        Edge Generation        Mode Shape Graph

Figure 2: Mode Shape Graph Construction: The process begins with node extraction from the FE model. Next, Node Reduction is performed to reduce the density of nodes from the input FE model, followed by Edge Generation to form the graph structure. Finally, Modal displacements in the x, y, and z directions, are assigned to each node as the features.

depending on the vibration mode. In this work, we focus on vehicle structures, as they provide a representative and challenging case for studying these issues.

Figure 2 summarises the process of constructing the mode shape graph dataset from FE simulation results. The graph nodes are sampled from the detailed FE model and the edges are generated using a $K$-nearest neighbors. The node features are defined by the modal displacement vectors associated with each vibration mode from the FE simulation results. Although mode shapes are derived from dynamic responses, each graph in this dataset is a static representation that encodes modal displacements as node features. Therefore, the task is framed as graph classification rather than temporal prediction. Overall, the dataset consists of 420 mode shape graphs, each with an average of 250 nodes and 4 edges per node, assigned to one of five mode shape classes (Figure 4).

2) **Baseline & Training Recipe:** The experimental setup for the vehicle mode shape dataset follows the same procedure as described in the benchmark experiments, including the choice of GNN architectures, baseline readout functions, and training configuration, with the hidden feature size fixed at 64 and the learning rate set to $10^{-3}$.

3) **Results:** As shown in Table 4, SUBRead achieves improvements over baseline GCN and GIN models and consistently outperforms other readout functions. In particular, we report results for $K = 4$ subgraphs, selected as the best-performing configuration from a sensitivity sweep over $K \in 2, 3, 4, 5$. This finding also highlights that SUBRead is not only effective on standard benchmarks but also adaptable to datasets developed from physical structures. For completeness, we compare SUBRead against hierarchical pooling methods, including node clustering approaches such as DiffPool (Ying et al., 2018) and MincutPool (Bianchi et al., 2020), and node dropping approaches such as TopK-Pool (Gao & Ji, 2019) and SAGPool (Lee et al., 2019b), where the pooling layer is placed between the second and third convolutional layers. Further implementation details are provided in the Appendix A.2. The results (Table 5) demonstrate that SUBRead consistently enhances classification performance compared to the baseline global sum readout for both GCN and GIN. SUBRead achieves strong improvements when combined with pooling methods such as SAGPool and MincutPool, and shows particularly strong synergy with GCN, highlighting its ability to complement simpler GNN architectures. For GIN, improvements are also observed, except when combined with DiffPool or MinCutPool, where no additional gains are achieved. Overall, the results confirm that SUBRead is a robust and complementary readout strategy.

4) **Qualitative Analysis**: To better understand how SUBRead makes predictions, we conduct a qualitative analysis of its learned representations. In particular, we visualise the sub-graphs highlighted by the alignment matrix ($K = 4$) when using GCN as the base architecture. These sub-graphs highlight the regions of the vehicle graph that contribute most to the graph-level representation prior to the final readout. Figure 3 shows that SUBRead segments the vehicle graph into coherent regions corresponding to meaningful physical parts. These regions are compared with clusters from other graph clustering methods and with partitions reflecting domain expert intuition. Among the clustering methods, SUBRead produces regions that are qualitatively aligned with expert-defined clusters, but we do not claim a one-to-one visual correspondence. In practice, NVH engineers typically focus on specific structural regions when classifying mode shapes (e.g., front versus rear, roof versus floor, or localised suspension regions), rather than relying on a single fixed or indescribable parti-

tion. SUBRead learns its clusters purely from the supervision features, and is therefore free to split or merge physical parts whenever this improves discrimination. As a result, the learned sub-graphs can differ from the simplified expert partition shown in Figure 3, but we observe that they repeatedly highlight similar functional regions (for example, the front left and right corners in torsion modes, where experts also concentrate their attention). This recurring focus on comparable structural areas, together with the performance gains in Table 5, suggests that SUBRead develops a form of structural awareness that is consistent with, but not identical to, human expert reasoning.

Table 4: Classification performance for vehicle mode shape dataset with different readout techniques (Figure 4)

| Readout | GNN Architecture | |
|---|---|---|
| | GCN | GIN |
| Global Sum (Baseline) | 90.6 (3.6) | 92.6 (2.6) |
| SortPool (0.6) | 94.9 (1.2) | 94.7 (2.4) |
| Set2Set (p=2) | 93.7 (3.3) | 94.8 (1.8) |
| Set2Set (p=4) | 95.0 (2.6) | 93.1 (3.2) |
| Attention | 70.7 (11.9) | 87.6 (6.2) |
| Covariance | 94.4 (2.1) | 93.6 (1.7) |
| SOPool$_{Bimap}$ | 91.6 (2.4) | 94.2 (2.3) |
| SOPool$_{attn}$ | 90.3 (3.4) | 92.2 (2.2) |
| GMT | 91.3 (3.4) | 93.8 (3.4) |
| Janossy-GRU | 90.3 (3.4) | 93.1 (2.2) |
| SUBRead (k=4) | **95.8** (1.1) | **94.9** (2.5) |

Table 5: Classification performance for vehicle mode shape dataset with different pooling techniques

| Pooling | GNN Architecture | |
|---|---|---|
| | GCN | GIN |
| Global Sum (Baseline) | 90.6 (3.6) | 92.6 (2.6) |
| SUBRead | **95.8** (1.1) | **94.9** (2.5) |
| DiffPool | 94.7 (2.4) | **94.2** (2.3) |
| DiffPool + SUBRead | **95.2** (2.6) | 94.2 (2.3) |
| MincutPool | 92.4 (2.9) | **95.1** (1.9) |
| MincutPool + SUBRead | **95.7** (2.3) | 94.6 (2.4) |
| TopK-Pool | 93.2 (2.0) | 94.4 (2.4) |
| TopK-Pool + SUBRead | **93.3** (3.0) | **94.7** (2.6) |
| SAGPool | 86.2 (6.5) | 67.6 (37.5) |
| SAGPool + SUBRead | **94.0** (2.7) | **93.6** (1.8) |

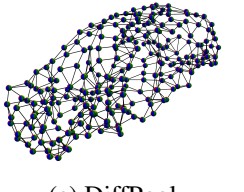 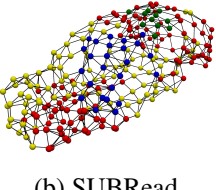 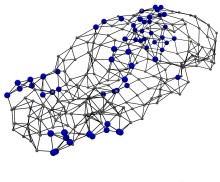 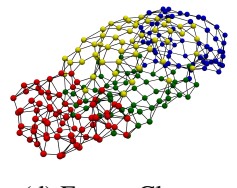

(a) DiffPool     (b) SUBRead     (c) TopKPool     (d) Expert Cluster

Figure 3: Comparison of structural clustering produced by different methods. (a) DiffPool performs soft clustering, allowing nodes to belong to multiple clusters. (b) SUBRead applies hard clustering, producing regions that align more closely with physical substructures. (c) TopK-Pool retains only the nodes with the highest importance scores. (d) Expert Cluster illustrates how domain experts intuitively partition the structure when classifying mode shapes.

## 5 CONCLUSION

We proposed SUBRead, an expressive readout function for GNNs that uses substructuring and attention to improve graph classification. By clustering nodes into sub-graphs, SUBRead can identify important local structures that reflect global graph properties, while attention weighting captures dependencies between sub-graphs. Our experiments on benchmark datasets show that SUBRead consistently outperforms existing readout functions. In practice, the method is particularly beneficial when the underlying graphs exhibit meaningful sub-structure, as the learned sub-graphs can align with functional or physical parts of the system; when such structure is less obvious, the number of sub-graphs $k$ can be treated as a tunable hyperparameter, and the resulting performance across different $k$ values provides a simple way to probe whether the data contain useful latent sub-structures. To further assess robustness, we evaluated SUBRead on a vehicle structural vibration dataset derived from a real-world automotive engineering problem, where it not only achieved high predictive performance but also revealed sub-graphs that were broadly aligned with meaningful physical components and consistent with interpretations made by domain experts. We believe that SUBRead provides GNN practitioners with a new perspective on the design of readout functions for graph classification at a time when interpretability is becoming increasingly critical for real-world applications, and that exploring adaptive strategies for choosing $k$ and scaling to even larger graphs are promising directions for future work.

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

## A APPENDIX

### A.1 DETAIL ON BENCHMARK DATASET

Table 6: Benchmark Dataset Statistics

| Dataset | # Samples | Classes | Average Nodes | Average Edges | Vertex Attributes | Vertex Labels |
|---|---|---|---|---|---|---|
| MUTAG | 188 | 2 | 17.93 | 19.79 | - | + |
| PROTEINS | 1,113 | 2 | 39.06 | 72.82 | +(1) | + |
| DD | 1,178 | 2 | 284.32 | 715.66 | - | + |
| ENZYMES | 600 | 6 | 32.63 | 62.14 | +(18) | + |
| Mutagenicity | 4,337 | 2 | 30.32 | 30.77 | - | + |
| NCI1 | 4,110 | 2 | 29.87 | 32.30 | - | + |
| COLLAB | 5000 | 3 | 74.49 | 2457.78 | - | - |
| REDDIT-Binary | 2000 | 2 | 429.63 | 497.75 | - | - |
| REDDIT-MULTI | 5000 | 5 | 508.52 | 594.87 | - | - |
| IMDB-Binary | 1000 | 2 | 19.77 | 96.53 | - | - |
| IMDB-MULTI | 1500 | 3 | 13 | 65.94 | - | - |
| Mode Shape (ours) | 420 | 5 | 250 | 4 | +(3) | - |

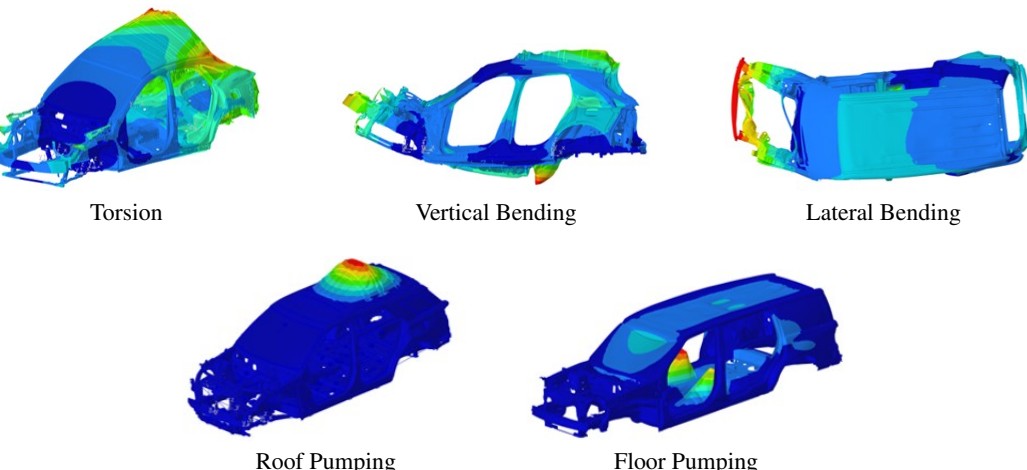

Figure 4: Mode Shape Label Examples. This vehicle mode shape dataset focuses on classifying global vehicle vibration responses observed at different resonant frequencies. The five characteristic classes are: (a) Torsion, (b) Vertical Bending, (c) Lateral Bending, (d) Roof Pumping, and (e) Floor Pumping.

1) **Datasets**: Table 6 summarises the key statistics of the TU benchmark datasets (Morris et al., 2020), including the number of graphs, average nodes, average edges, node attributes, and node labels. For bioinformatics, MUTAG (Debnath et al., 1991) is a dataset consisting mutagenic aromatic and heteroaromatic nitro compounds. The goal is to predict the mutagenicity (a binary classification) of the compounds. PROTEINS (Dobson & Doig, 2003) and D&D (Dobson & Doig, 2003) datasets consist of proteins where each protein is model as a graph. Both dataset share the same task which is to predict whether protein is enzyme or not. ENZYMES (Schomburg et al., 2004) consists of protein tertiary structures which can be categorised into six Enzyme Commission (EC) top-level classes where the task is to predict the type of each enzyme. Mutagenicity (Kazius et al., 2005) dataset task is to identify which chemical compound can cause mutation. NCI1 (Wale et al., 2008) dataset is a part of the National Cancer Institute (NCI) datasets. The classification task is to predict the biological activity of the compound against a specific type of cancer.

For social-network, COLLAB (Yanardag & Vishwanathan, 2015) is related to scientific collaboration dataset where the task is to determine whether the ego-collaboration belongs to High Energy,

Condensed Matter or Astro Physics field. IMDB-BINARY (Yanardag & Vishwanathan, 2015) and IMDB-MULTI (Yanardag & Vishwanathan, 2015) datasets are related to generes movies, binary dataset includes Romance and Action genere whereas multiclass dataset consists of Comedy, Romance and Sci-Fi genres. the task is to identify the genre of a given graph which collected actor/actress and genre information of different movies on IMDB. REDDIT-BINARY (Yanardag & Vishwanathan, 2015) and REDDIT-MULTI (Yanardag & Vishwanathan, 2015). The binary dataset corresponds to an online discussion thread where the task is to predict whether a given graph belongs to question/answer-based community or a discussion-based community. For multiclass dataset, the task is to identify what the class of subreddit for a graph including worldnews, videos, AdviceAnimals, aww and mildlyinteresting.

For the Mode Shape dataset, the task is to classify graphs into one of five mode shape categories: Torsion, Vertical Bending, Lateral Bending, Roof Pumping, and Floor Pumping. The dataset was constructed from mode shapes of various SUV models, with labels assigned by domain experts to ensure reliability. Representative samples are shown in Figure 4.

## A.2 ADDITIONAL INFORMATION ON EXPERIMENTAL SETUP

**Baselines:** To validate the effectiveness of our SUBRead layer compared to existing readout layers, we tested it using two GNN architectures including Graph Convolutional Networks (GCNs), and Graph Isomorphism Networks (GINs). By incorporating the SUBRead layer into these different architectures, we will analyse the performance across several benchmark datasets. The following summarises the details of each GNN variant:

- **GCN:** (Kipf & Welling, 2016) This architecture extends the concept of convolutional neural networks to graph-structured data. As a pioneer in the field of graph convolution, the approach is computationally efficient, avoiding the explicit computation. GCNs sometimes outperform more advanced GNN models, demonstrating their robustness and versatility (Dwivedi et al., 2023). This model serves as the foundation for many GNN variants, such as GIN, GraphSAGE, and GAT.
- **GIN** (Xu et al., 2018): A mathematically grounded model that shows sum aggregation, when combined with a multilayer perceptron (MLP), is more powerful than other global operations including max and mean. This design makes GIN as expressive as the Weisfeiler–Lehman test and effective at capturing complex graph structures.

**Readout Setup:** For the readout experiments, we fixed the number of convolution layers to three across all GNN architectures, followed by the readout function and a classification layer. The explanation of each method and the selection of its hyperparameters are described below.

1) **Set2Set** (Vinyals et al., 2015) is the one of the earlist methods to produce expressive readout function. The concept was originally proposed to address learning on unordered sets using an LSTM with attention to produce permutation-invariant embeddings, and was later adopted by Gilmer et al. (2017) as a readout function in GNNs. We report results for $p = 2$ and $p = 4$, where $p$ specifies the number of LSTM-based aggregation steps in Set2Set.

2) **SortPool** (Zhang et al., 2018) generates a graph level representation from a structural score inspired by Weisfeiler & Leman (1968) refinement method. After a GNN computes node features, the nodes are ranked according to a structural score. The top-k nodes are selected, forming a fixed-size matrix that is then processed by 1D convolutional and pooling layers. Following the original work, we fixed the pooling size at 60%.

3) **Attention** (Girdhar & Ramanan, 2017; Li et al., 2019) introduces a learnable function which captures the pairwise interactions of features. This allows the model to assign weights to spatial descriptors and focus on the most discriminative regions.

4) **Covariance** (Wang et al., 2020) has been widely explored in image tasks such as image categorization, facial expression recognition, and texture classification. Unlike mean or max pooling, which only capture first-order statistics, covariance pooling encodes second-order information by modeling the pairwise interactions of features.

5) **SOPool** (Wang et al., 2020) improves covariance pooling by proposing two variants: $\text{SOPool}_{Bimap}$ and $\text{SOPool}_{Attn}$. The Bimap variant applies bilinear mapping to the covariance ma-

trix, projecting it into a lower-dimensional space while preserving second-order statistics, while the Attn variant map the covariance matrix into a single vectors through linear mapping function. Both approaches aim to retain the benefits of second-order pooling while making it more efficient and robust.

6) **GMT** (Baek et al., 2021) formulates graph readout as a set encoding task, where learnable seed vectors interact with node embeddings through multi-head attention. This design allows the model to capture diverse aspects of the graph and produce expressive graph-level representations. For consistency, we used four seed vectors across all experimental settings.

**Pooling Setup:** We follow a standard GNN setup with three convolution layers, where pooling is applied between the second and last layer. Graph pooling methods are commonly divided into two categories: hierarchical pooling, which progressively coarsens the graph by clustering nodes (e.g., DiffPool and MinCutPool), and node-dropping approaches, which reduce the graph size by discarding nodes according to a scoring function (e.g., TopK and SAGPool). While pooling and readout operations are similar, their objectives differ: pooling generates a smaller but still structured graph for subsequent layers, whereas readout collapses the entire graph into a single representation for downstream tasks such as graph classification. For consistency, we set the dropping ratio of all pooling methods to 0.25 and used sum operation as the global readout after third layer. The explanation of each method in are described below.

1) **Diffpool** (Ying et al., 2018) introduces a soft cluster assignment that maps nodes into clusters, with cluster quality guided by auxiliary objectives such as link prediction and entropy regularisation.

2) **MincutPool** (Bianchi et al., 2020) follows a similar clustering strategy with Diffpool but replaces the auxiliary losses with a min-cut objective along with an orthogonality constraint to ensure balanced clusters.

3) **TopKPool** (Gao & Ji, 2019), proposed as part of Graph U-Net, assigns a score to each node based on a projection of its features and retains only the top fraction of nodes.

4) **SAGPool** (Lee et al., 2019b) extends TopKPool by computing node scores by graph convolution, making it more structure-aware. Unlike TopKPool, which relies solely on node features, SAGPool incorporates both features and neighborhood information when selecting nodes.

### A.3 USE OF LANGUAGE MODELS

The authors declare that large language models (LLMs) were used solely to polish grammar and improve clarity of writing. All research ideas, methodologies, analyses, and conclusions presented in this paper were fully conceived and developed by the authors.

