# OpenReview forum: "SUBRead: Clustering Sub-Graphs for Graph-Level Readout"
_ICLR.cc/2026/Conference — Submitted to ICLR 2026_

### Official Review · Reviewer_zozn · 2025-10-29

**Soundness:** 2
**Presentation:** 1
**Contribution:** 2
**Rating:** 2
**Confidence:** 5

**Summary:**

This paper proposes SUBRead, a graph neural network readout function that clusters nodes into subgraphs and integrates attention mechanisms to generate graph-level representations. The method outperforms baseline methods on multiple benchmark datasets from bioinformatics and social networks, and further demonstrates its effectiveness on a real-world vehicle structural vibration dataset.

**Strengths:**

The method is evaluated on several benchmark datasets (PROTEINS, MUTAG, NCI1) and a real-world engineering dataset, providing evidence of its effectiveness.

**Weaknesses:**

1. The paper demonstrates limited novelty, as graph pooling operators beyond simple average or sum pooling has already been studied extensively in the literature. in particular, sub-graph clustering and attention based graph pooling has been explored in the reference [1], and its differences with the proposed method should be discussed.

2. The comparison with existing methods appears limited. It is recommended that the authors include additional baselines using more advanced readout techniques (e.g., Janossy GRU/MLP [2], kerRead[3]) to better demonstrate the effectiveness of the proposed approach.

3. The paper lacks sufficient parameter sensitivity and ablation studies, making it difficult to assess the contribution of each component in the framework.

4 The writing of the paper and the figures should be improved. Many technical details and key motivations are missing.


[1] Gaoqi He, Shun Liu, Kai Zhang, Honglin Li. Prototype-based Contrastive Substructure Identification for Molecular Property Prediction. Briefings in Bioinformatics

[2] Buterez, David, et al. "Graph neural networks with adaptive readouts." Advances in Neural Information Processing Systems 35 (2022): 19746-19758.

[3] Yu, Jiajun, et al. "Kernel readout for graph neural networks." Proceedings of the Thirty-Third International Joint Conference on Artificial Intelligence, IJCAI-24. 2024.

**Questions:**

(1) the subgraph clustering step should guarantee that the segmentation of the initial graph should be physically feasible, i.e., avoiding isolated node, subgraph sizes should be balanced, etc., which requires considering both similarity-between sub-graphs and topological relations, which this is largely ignored.

(2) The paper uses learnable prototypes to perform subgraph clustering of nodes. Could the authors provide visualizations or other analyses to demonstrate the relationship between the clustering results and the underlying graph structure, thereby providing stronger evidence for the effectiveness of the method?

(3) how do you choose the number of clusters, k in practice? do you need to used validation set to tune this hyper-parameter?

(3) is the equation in line 243 a self-attention?

---

### Official Review · Reviewer_kD2A · 2025-10-30

**Soundness:** 2
**Presentation:** 2
**Contribution:** 2
**Rating:** 2
**Confidence:** 3

**Summary:**

This paper proposes SUBRead, a novel graph readout function for Graph Neural Networks (GNNs). Instead of aggregating node representations using simple global pooling operations (e.g., sum/mean/max), SUBRead first clusters a graph into local subgraphs, then applies subgraph-level aggregation, followed by an attention mechanism to combine these subgraph embeddings into a final graph-level representation. The method is fully differentiable and can be integrated with standard GNN architectures. The authors evaluate SUBRead on benchmark datasets as well as a real-world structural vibration classification problem, where graphs share identical topology but differ substantially in node features. Across experiments, SUBRead outperforms existing readout techniques and yields interpretable substructures aligned with expert reasoning.

**Strengths:**

1. Novel Readout Perspective: The focus on subgraph-level aggregation is a meaningful and well-motivated departure from commonly used global pooling strategies, addressing known limitations of over-compressive readouts.

2. Strong Empirical Results: SUBRead demonstrates consistent improvements across benchmarks and shows particular effectiveness in the mode shape classification task, where structural interpretability matters.

3. Interpretability Component: The revealed substructures are not merely performance-driven—they appear aligned with domain insights, which is valuable for real-world engineering applications.

**Weaknesses:**

1. Clustering Method Justification: The choice of clustering mechanism could be explained more rigorously. It is not fully clear how sensitive the performance is to the number of clusters or selection of the clustering algorithm.

2. Computational Complexity: The added clustering and attention layers introduce overhead relative to simpler readouts. The paper would benefit from a more explicit complexity analysis.

3. Limited Analysis of Failure Cases: The paper focuses on positive performance outcomes but provides little discussion of where SUBRead might underperform (e.g., graphs with no meaningful substructure).

**Questions:**

1. How should practitioners choose or tune the number of clusters? Is there a heuristic or learning mechanism planned for future work?

2. How does SUBRead behave when the graph size varies significantly across samples? Does clustering remain stable?

---

### Official Review · Reviewer_VFtz · 2025-10-31

**Soundness:** 3
**Presentation:** 3
**Contribution:** 2
**Rating:** 4
**Confidence:** 4

**Summary:**

The paper proposes a novel readout operation for graph neural networks. Similar to related works, it considers learnable prototype vectors. But instead of using them to attend to the entire graph, they use them to compute "subgraphs" (nodes clustered based on similarity to the prototypes) and aggregate the subgraphs using attention. The concatenated output vectors (i.e., one per subgraph) represent the graph-level representation.

**Strengths:**

- The paper is well-written and organized
- The approach sounds interesting and novel (but the related work discussion seems to be lacking, see below).
- The qualitative analysis for the vehicle data is very interesting and could be expanded to other datasets.

**Weaknesses:**

- The paper misses to discuss/compare to subgraph-related works, where the subgraphs are applied during message passing (e.g., [1], [2]). Given that the authors perform many experiments over molecular data, it would be good to understand similarities or differences between these approaches.  There also seems to be a connection to virtual nodes, which might be interesting to discuss.

- My main criticism is the empirical evaluation. The baselines in Tables 1&2 are rather simple. The really interesting numbers on improving existing pooling methods, presented thereafter, are only given over a single dataset. So it's open if this generalizes.


[1] Frasca et al. Understanding and Extending Subgraph GNNs by Rethinking Their Symmetries. Neurips'22.

[2] Lee et al. Graph Convolutional Networks with Motif-based Attention. CIKM'19.

**Questions:**

---------------------------------

---

### Official Review · Reviewer_ijdj · 2025-11-03

**Soundness:** 2
**Presentation:** 3
**Contribution:** 2
**Rating:** 4
**Confidence:** 3

**Summary:**

The paper proposes SUBRead, a graph readout that (i) assigns nodes to one of k learned centroids via distance, producing subgraphs, (ii) aggregates each subgraph, (iii) applies self-attention across the subgraph vectors, and (iv) concatenates them as the final graph embedding. Experiments span several TU datasets and a small real-world “vehicle mode-shape” dataset. The idea is simple and potentially useful, but there are some concerns around differentiability of the clustering step, permutation invariance of the readout, fairness/strength of baselines, statistical rigor, and scaling.

**Strengths:**

The authors propose a straightforward and modular approach that could improve on existing graph readout methods and easily drop into existing GNNs.

The authors compare against a number of reasonable baselines across two architectures (GCN/GIN) and a number of readouts (Sort-Pool, Set2Set, Attention, Covariance, SOPool, and GMT).

They compare over 11 datasets from the TU datasets collection and one small real world vehicle mode shape classification dataset.

**Weaknesses:**

The method forms a binary assignment matrix over distances to learned centroids (hard, 0/1). The text references N3Net to motivate a “learnable relaxation,” but the presented rule is a hard winner-take-all mapping (argmin to one-hot), which as described seems non-differentiable. It is thus currently unclear how gradients propagate from the loss to node assignments and centroids e.g., via straight-through estimator, Gumbel-softmax/concrete? Please clarify.

What is the time/memory overhead vs. baselines? Complexity appears O(nk) for the distance matrix plus O(k^2) attention... not large for k≤5 but results on larger graphs/benchmarks are absent. Please report wall-clock, peak memory, and parameter counts, and include a large-graph stress test.

Are we supposed to compare the visual clusters in Fig. 3 and find the SUBRead and Expert Cluster similar? They do not look all that similar to me... am I to just believe the authors these are 'meaningful physical parts'? Some more description (and better quantification) would be apprecaited here.

The authors seem to neglect some recent relevant literature on related learnable graph methods (e.g., https://neurips.cc/virtual/2024/poster/94335) and all
their baselines are from 2020 or before, potentially neglecting some like:
https://arxiv.org/abs/2209.07817
https://www.ijcai.org/proceedings/2024/0277.pdf
https://www.nature.com/articles/s41467-025-60252-z

**Questions:**

Concatenating the attended subgraph vectors makes the final embedding depend on the ordering of clusters... is this an issue?

Results sweep k∈{2,3,4,5} and sometimes pick best-k, while several baselines are fixed (e.g., GMT seeds fixed to 4). Please confirm that baseline hyperparameters were comparably tuned/matched to avoid search budget asymmetry.

Are the OOM results unavoidable? Please provide per-method memory settings or explain why you can't scale to a common feasible setting.

---

> ### Author Response · Authors · 2025-12-02
>
> **5) Concatenated order**
>
>
> In SUBRead, after clustering we obtain Ksubgraph vectors, which are then concatenated into a single graph representation. If one were to manually permute these Kslots after training, the resulting embedding would indeed change. However, this dependence is only on the internal index of each prototype, not on the ordering of nodes in the input graph.
> The prototype indices are tied to learnable parameters; once training begins, their order is fixed, and permuting them would be analogous to permuting channels in a hidden layer of a standard neural network. This does not affect permutation invariance with respect to node permutations: permuting the node order changes only the ordering of node embeddings, not the assignment of nodes to prototypes, so each subgraph vector still aggregates over the same subset of nodes. This explanation has been added in Section 3.3 to further improve the clarity of the algorithm.
>
>
> **6) How to choose best K**
>
>
> We agree that guidance on choosing Kis important. In practice, we use two complementary strategies:
> (i) Domain-informed choice. When the data contains a natural decomposition (e.g., functional regions of an engineering structure, coarse molecular fragments), practitioners can set Kto reflect this prior structure. This is the setting for our engineering mode-shape dataset.
> (ii) Validation-based tuning. For datasets without clear physical substructures (e.g., social-network graphs), we simply sweep a small range $K \in \\{2,3,4,5\\} $ and select the value that yields the best validation performance.
> In our experiments, we follow strategy (ii) on the TU benchmarks. The results (highlighted in Section 4.2, Table 4) show that SUBRead is relatively stable across this range, with a mild preference for $K=4$.
>
> **7) Explain why cannot avoid OOM**
>
> REDDIT-BINARY contains substantially larger graphs than the other benchmarks. The covariance-pooling baseline requires computing dense second-order statistics, whose memory cost grows quadratically with the feature dimension and linearly with graph size. On our local hardwae (NVIDIA RTX A1000, 32 GB RAM), this combination leads to out-of-memory errors even with very small batch sizes, whereas all other methods (including SUBRead) train without issues under the same settings.
> In principle, the covariance baseline could be evaluated by using more aggressive downsampling, smaller feature dimensions, or higher-memory hardware, but these options are beyond our current computational budget. We therefore mark the corresponding entries as “OOM” in the table and leave some of the readout methods on the REDDIT dataset for future investigation.

---

### Meta-Review · Area_Chair_5WF4 · 2025-12-31

**Summary:**

This paper proposes SUBRead, a subgraph-based readout function for graph-level representation learning. While the idea is intuitive, the contribution is limited and closely related to existing work on prototype-based, attention-based, and subgraph-aware readouts. Multiple reviewers raised concerns about novelty, empirical rigor, baseline coverage, and clarity. Importantly, the authors only provided substantive rebuttal to one reviewer, leaving several major concerns from other reviewers unaddressed. Overall, the paper does not meet the acceptance bar.

**Reviewer Concerns:**

Overall, the concerns from reviewers were not adequately addressed in the rebuttal.

Several reviewers raised fundamental concerns regarding the novelty of the proposed readout. Similar ideas combining clustering, prototypes, attention, or subgraph aggregation have been studied extensively in prior work. The reviewers noted that the paper does not clearly differentiate SUBRead from these existing methods, especially more recent approaches. The related work discussion and positioning were considered incomplete.

There were also significant concerns about the empirical evaluation, including limited and weak baseline comparisons, unclear fairness in hyperparameter tuning, lack of sufficient ablation and sensitivity analysis, and limited evidence that the reported improvements generalize beyond a small subset of datasets.

In addition, reviewers pointed out technical issues such as permutation dependence in the concatenation-based readout, unclear clustering behavior, computational overhead, scalability to larger graphs, and missing analysis of failure cases. These issues affect both the soundness and practical applicability of the method.

Crucially, the authors only provided detailed rebuttal responses to one reviewer. Concerns raised by several other reviewers were not substantively addressed, and no new experiments were provided to resolve these points. As a result, the rebuttal did not lead to meaningful convergence among reviewers.

**Reviewer Scores:**

Reviewer ijdj: Rated the paper slightly below the acceptance threshold (4). While some clarification was provided, major concerns about novelty, evaluation rigor, and scalability remain. The score would likely remain 4.

Reviewer VFtz: Gave a below-threshold score (4), mainly due to limited empirical validation and missing comparisons. These concerns were not directly addressed since the authors did not provide a response. The score would remain 4.

Reviewer kD2A: Gave a clear reject (2), citing limited novelty, insufficient analysis, and missing complexity discussion. No rebuttal was provided to address these points. The score would remain 2.

Reviewer zozn: Gave a strong reject (2) with high confidence, emphasizing lack of novelty, missing baselines, and poor presentation. These concerns were not addressed in the rebuttal as no response was provided. The score would remain 2.

---

### Decision · Program_Chairs · 2026-01-26

Reject